# Relationship between Birth Order and Postnatal Growth until 4 Years of Age: The Japan Environment and Children’s Study

**DOI:** 10.3390/children10030557

**Published:** 2023-03-15

**Authors:** Aya Yoshida, Kayo Kaneko, Kohei Aoyama, Naoya Yamaguchi, Atsushi Suzuki, Sayaka Kato, Takeshi Ebara, Mayumi Sugiura-Ogasawara, Michihiro Kamijima, Shinji Saitoh

**Affiliations:** 1Department of Pediatrics and Neonatology, Graduate School of Medical Sciences, Nagoya City University, Nagoya 4670001, Japan; 2Department of Occupational and Environmental Health, Graduate School of Medical Sciences, Nagoya City University, Nagoya 4670001, Japan; 3Department of Ergonomics, Institute of Industrial Ecological Sciences, University of Occupational and Environmental Health, Kitakyushu 8078555, Japan; 4Department of Obstetrics and Gynecology, Graduate School of Medical Sciences, Nagoya City University, Nagoya 4670001, Japan

**Keywords:** birth order, early childhood, height, prospective cohort study, short stature

## Abstract

Later-borns tend to be shorter than first-borns in childhood and adulthood. However, large-scale prospective studies examining growth during infancy according to birth order are limited. We aimed to investigate the relationship between birth order and growth during the first 4 years of life in a Japanese prospective birth cohort study. A total of 26,249 full-term singleton births were targeted. General linear and multivariable logistic regression models were performed and adjusted for birth weight, parents’ heights, maternal age at delivery, gestational weight gain, maternal smoking and alcohol drinking status during pregnancy, household income, breastfeeding status, and Study Areas. The multivariate adjusted mean length Z-scores in “first-borns having no sibling”, “first-borns having siblings”, “second-borns”, and “third-borns or more” were −0.026, −0.013, 0.136, and 0.120 at birth and −0.324, −0.330, −0.466, and −0.569 at 10 months, respectively. Results similar to those at 10 months were observed at 1.5, 3, and 4 years. The adjusted odds ratios (95% confidence intervals) of short stature at 4 years in “first-borns having siblings”, “second-borns”, and “third-borns or more” were 1.08 (0.84–1.39), 1.36 (1.13–1.62), and 1.50 (1.20–1.88), respectively, versus “first-borns having no sibling”. Birth order was significantly associated with postnatal growth and may be a factor predisposing to short stature in early childhood.

## 1. Introduction

In Japan, the mean height of adults has been gradually decreasing for three decades [1,2]. As those with a short stature have a higher risk of intellectual and psychological disabilities [3,4], coronary heart disease and stroke [5,6,7], and pregnancy disorders [8,9,10], it is important to elucidate the determinants and biological factors regulating postnatal growth. Numerous studies have previously reported that postnatal growth is affected by genetic factors, represented by the parent’s heights; maternal nutritional condition during pregnancy; birth weight [2]; domestic environment [11,12], including household income; and nutritional intake during infancy [13,14,15].

Several studies have indicated that birth order is negatively associated with postnatal growth. Previous cross-sectional studies have reported that the height of first-borns is greater than that of later-borns in both early childhood [16,17] and adulthood [18,19,20].

Other studies reported that later-borns were shorter than first-borns at 6 months [21] and 12 months [22] or older, even though the birth length of later-borns tended to be greater than that of first-borns. However, it is inconclusive from which period the significant negative association between birth order and postnatal growth can be observed, as previous studies did not have enough information on length during infancy. Additionally, to the best of our knowledge, no study has examined the relationship between birth order and short stature in early childhood. A previous study reported a significant positive correlation between height in early childhood and adulthood [23], and, thus, an assessment of factors related to early childhood height, such as birth order, may provide useful knowledge to predict future short stature.

Several researchers have considered that the nutritional disadvantages in infancy, caused by unfairly divided family resources, in later-borns can result in a negative association between birth order and postnatal growth [12,16,24]. If the domestic environment, including infancy nutritional condition and the presence of siblings, truly affects postnatal growth, first-borns with no siblings should be taller than first-borns with siblings.

Therefore, this study aimed to investigate the hypothesis that, even in Japan, later-borns are born larger than first-borns, but growth reverses during early childhood, which may be a risk factor for short stature. To this end, we examined the relationship between birth order (considering the existence of a sibling for first-borns) and postnatal growth using a Z-score of height at birth and at 10 months, 1.5 years, 3 years, and 4 years of age and the association between birth order and short stature among 4-year-old children, using large datasets from ongoing multicenter prospective birth cohort studies across Japan.

## 2. Materials and Methods

The Japan Environment and Children’s Study (JECS) is an ongoing nationwide, multicenter, prospective birth cohort study in Japan, which examines the effects of environmental factors and medical, psychosocial, and lifestyle conditions during pregnancy and childhood on the infant’s health and development at birth and later in life. The details of the study design have been described elsewhere [25]. The JECS was established in 2011 as a collaboration among the Program Office (National Institute for Environmental Studies), the Medical Support Center (National Center for Child Health and Development), and 15 Regional Centers (Hokkaido, Miyagi, Fukushima, Chiba, Kanagawa, Koshin, Toyama, Aichi, Kyoto, Osaka, Hyogo, Tottori, Kochi, Fukuoka, and South Kyushu/Okinawa) [25,26]. Briefly, between January 2011 and March 2014, pregnant women were recruited during their first trimester in the Co-operating health care providers and/or local government offices that issue Maternal and Child Health Handbooks. The handbook is issued by all local governments in Japan to all expectant and nursing mothers in order for them to receive public services related to pregnancy, childbirth, and childcare and to keep at home with the results of periodical checkups. The criteria for eligible participants were (i) living in the Study Area in one of the 15 Regional Centers at the time of recruitment and expected to continue to reside in Japan for the foreseeable future; (ii) an expected delivery between 1 August 2011 and mid-2014; and (iii) an understanding of the Japanese language so that they may be able to answer the self-administered questionnaire.

The JECS protocol was reviewed and approved by the Ministry of the Environment’s Institutional Review Board on Epidemiological Studies (No. 100910001, 6 April 2010) and the Ethics Committees of all participating institutions. Written informed consent was obtained from all participants. For this study, we used the dataset released in February 2022, which includes child growth information up to 4 years of age (dataset jecs-qa-20210401).

### 2.1. Study Participants

From January 2011 to March 2014, 104,059 fetal records were registered in the JECS. Multiple births, preterm infants, and post-term infants were excluded because their growth patterns differ from those of term singletons [27,28]. The first 100,300 live births were extracted. Multiple pregnancy (*n* = 1891), delivery before 37 weeks (*n* = 4617), delivery after 42 weeks (*n* = 226), missing data for gestational age at delivery (*n* = 289), and two or more enrolled pregnancies to the same mother (*n* = 5195) were excluded, and 88,082 mother and infant pairs were extracted. Subsequently, the following cases were excluded: (1) 56,855 cases with missing and conflicting data for children’s length/height, such as ±5.0 standard deviation (SD), including birth length (*n* = 268), length at 10 months (*n* = 29,390), length at 1.5 years (*n* = 14,559), height at 3 years (*n* = 7803), and height at 4 years (*n* = 4835); (2) 4655 cases with missing and conflicting data for explanatory variables and covariates, including birth order (*n* = 289), parity (*n* = 897), the height of the mother (*n* = 2), the height of the father (*n* = 829), weight gain during pregnancy (*n* = 608), smoking status during pregnancy (*n* = 181), drinking status during pregnancy (*n* = 171), household income (*n* = 1467), and breastfeeding status at 2 years (*n* = 221); and (3) children with congenital cerebral abnormalities (*n* = 24), congenital cardiac abnormalities (*n* = 275), and chromosomal abnormalities (*n* = 14). Finally, a total of 26,249 mother and child pairs of singleton births at term were eligible for the study (Figure 1). 

We performed a sensitivity analysis of the exclusion procedure and found no significant differences in the sex or birth weight/height z-score of children between all JECS participants [26] and participants eligible for the present study. However, the maternal age at delivery and the proportion of first-borns in the current analysis were significantly higher than those among all JECS participants with singleton live births (*p* < 0.05).

### 2.2. Birth Order

The birth order of the participants was reported in a self-administered questionnaire by mothers or caregivers 2.5 years after childbirth. The questionnaire asked questions regarding the presence and number of younger/older siblings living with the study participant. We classified participants into four groups: “first-borns who had no sibling”, “first-borns who had siblings”, “second-borns”, and “third-borns or more”. For some analyses, we renamed these four groups as “Group IA”, “Group IB”, “Group II”, and “Group III”, respectively, to simplify the detailed inter-group comparisons.

### 2.3. Z-Score of Length/Height and Short Stature

Birth length was transcribed from the medical records of physicians, mid-wives/nurses, and/or research coordinators. Mothers or caregivers reported their children’s most recent length/height and the date of the measurement in questionnaires distributed regularly twice per year. Although the ages at the time of the reported measurements varied, only those within plus or minus two months of the aforementioned baseline ages were taken, converted to Z-scores, and then tabulated separately for each baseline age. The mean ages at measurements of length/height were 10.4 ± 0.8, 17.8 ± 0.8, 35.7 ± 0.8, and 47.8 ± 0.8 months. The Z-scores of length/height were calculated according to the growth standard charts for Japanese children using the lambda-mu-sigma (LMS) method, with the closest LMS values for each measurement age used [29,30]. Short stature at 4 years of age was defined as less than −2 SD.

The Japanese Maternal and Child Health Act obligates municipalities to issue all pregnant women a Maternal and Child Health Handbook for them to keep their medical information in. The handbook has pages for recording children’s weight and length/height measured at medical institutions or local government offices, as well as the date of the measurement. The act also obligates municipalities to implement free checkups for children at 1.5 and 3 years of age, and 72% of municipalities voluntarily provide additional checkups for 9- to 10-month-old children. The infant/child health checkup coverage at 9 to 10 months, 1.5 years, and 3 years is high at 85.7%, 95.7%, and 94.6%, respectively. Furthermore, the “Manual for the evaluation of infant and child physical development” reports that children’s height at 2 years or more must be measured by health care professionals in the standing position to the closest 0.1 cm [31]. Therefore, the self-reported measurements, which are mostly transcriptions from the Maternal and Child Health Handbook by mothers or caregivers, are considered to have a certain level of reliability. Additionally, we made sure that the distributions of length/height in the current study at each measurement point were similar to the results from the Japanese national survey [29].

### 2.4. Covariates

Socio-economic information and lifestyle-related data were self-reported during early and/or mid-pregnancies [32]. The sex of the child, gestational age at birth, and data regarding neonatal anthropometric measurements were transcribed from the medical records on childbirth by the Co-operating health care providers. The gestational age in weeks and days was determined by carrying out an ultrasound examination of the first trimester and/or an estimation from the last menstrual period. The Z-score of the birth weight for gestational age was calculated based on Japanese neonatal anthropometric charts. Subsequently, we defined the Z-scores of the birth weight for gestational age of less than −1.28155 (falling below the 10 percentile range) as small for gestational age (SGA), from −1.28155 to 1.28155 as appropriate for gestational age (AGA), and greater than 1.28155 (90 percentile range or higher) as large for gestational age (LGA) [30] in the stratified analysis. Breastfeeding status at 2 years of age was reported by mothers or caregivers. The father’s height was self-reported or reported by the partner. 

Maternal age at delivery [33], parents’ heights [34], weight gain during pregnancy [35], the Z-scores of the birth weight [2], the sex of the child, maternal smoking status [36], maternal alcohol drinking status [37], household income [38], and breastfeeding status at 2 years [39] were used as covariates in the present study, as they have been reported to be factors affecting child growth. Maternal age at delivery, parents’ heights, weight gain during pregnancy, and the Z-scores of the birth weight were used as continuous variables. 

The sex of the child (male, female) was dichotomized. Maternal smoking status during pregnancy was classified as “Never”, “Previously did, but quit before recognizing current pregnancy”, “Previously did, but quit after finding out current pregnancy”, and “Yes, I still smoke”. Alcohol drinking status during pregnancy was classified as “never drinker”, “former drinker”, and “current drinker”. Household income levels (million JPY) were divided into less than 2, 2 to 6, 6 to 10, and 10 or greater. Breastfeeding status at 2 years was classified as “currently breastfeeding”, “stopped breastfeeding before 2 years”, and “never breastfed”. The Study Areas were divided into the locations of the 15 Regional Centers, which included “Hokkaido”, “Miyagi”, “Fukushima”, “Chiba”, “Kanagawa”, “Koshin”, “Toyama”, “Aichi”, “Kyoto”, “Osaka”, “Hyogo”, “Tottori”, “Kochi”, “Fukuoka”, and “South Kyushu/Okinawa”.

These covariates were chosen based on recent biological and epidemiological insights regarding child growth or a strategy of selecting covariates according to a certain level of statistical significance (e.g., *p* < 0.1 or *p* < 0.05) in the univariate analyses. Therefore, we concluded that a fully adjusted model reasonably well discriminated short stature at 4 years, with a rate of correct classifications of 90.2% and a *p*-value of 0.55 for goodness-of-fit using the Hosmer–Lemeshow test.

### 2.5. Statistical Analysis

To compare the adjusted means of the Z-score of length/height at birth and at 10 months, 1.5 years, 3 years, and 4 years of age according to the four birth order groups, multivariable general linear models were performed with post hoc Bonferroni tests. The *p*-value for the linear trend was obtained using the polynomial method in the general linear model. The odds ratios (ORs) and 95% confidence intervals (CIs) of short stature at 4 years of age according to birth order were estimated using multivariable-adjusted logistic regression models. Model 1 was adjusted for the age of the mother at delivery. Model 2 was adjusted for the variable in model 1 and the Z-score of birth weight. Model 3 was the same as model 2 but with the parent’s height added as an additional variable. Model 4 was the same as model 3 but with weight gain during pregnancy, maternal smoking and drinking status during pregnancy, household income, breastfeeding at 2 years of age, and Study Areas (Hokkaido, Miyagi, Fukushima, Chiba, Kanagawa, Koshin, Toyama, Aichi, Kyoto, Osaka, Hyogo, Tottori, Kochi, Fukuoka, and South Kyushu/Okinawa) included as variables. Since postnatal growth patterns, such as catch-up and catch-down growth, may be influenced by the birth weight for gestational age, we further performed a stratified analysis by SGA, AGA, and LGA.

All statistical analyses were performed using IBM SPSS Statistics for Windows software version 28 (IBM Corp.: Armonk, NY, USA). A *p*-value of less than 0.05 was considered significant.

## 3. Results

### 3.1. Participant Characteristics

A total of 26,249 children born at term were eligible for the present study: 9657 (36.8%) first-borns who had no siblings, 3463 (13.2%) first-borns who had siblings, 9240 (35.2%) second-borns, and 3889 (14.8%) third-borns or more. The study participants had a mean (SD) gestational age at birth of 39.07 (±1.14) weeks, and 50.7% were males. The Z-score of birth weight and length and the age of the mother at delivery were the highest in the third-borns or more. Maternal weight gain during pregnancy was higher in the group of first-borns than in the other groups. The proportion of current smoking and drinking status during pregnancy in first-borns was lower than that in the other groups. Household income tended to be lower in the groups of second-borns and third-borns or more. Regarding the breastfeeding status at 2 years, the proportion of those who were currently breastfeeding was higher in the group of third-borns or more than in the other groups. Significant differences were not observed in maternal height according to the birth order groups. A significantly higher paternal height was observed in the group of second-borns (Table 1).

### 3.2. Adjusted Mean Z-Score of Length/Height from Birth to 4 Years According to Birth Order

Significant differences in the Z-score of length/height were not observed between first-borns with and without siblings from birth to 4 years of age. Third-borns or more indicated a gradually significantly smaller adjusted Z-score of length/height compared with first-borns and second-borns (*p* < 0.05; *p* for linear trend < 0.01), except at birth. A post hoc test using the Bonferroni test indicated no significant differences between second-borns and third-borns at birth and 1.5 years of age (Table 2). In addition, similar results were observed in the analysis limited to only males or females. From another perspective, the degree of height difference in birth order was narrow until 1.5 years of age, but it slightly widened at 3 and 4 years of age.

### 3.3. Association between Birth Order and Short Stature at 4 Years of Age

A total of 792 (3.0%) children with a short stature were observed at 4 years of age. Although the proportion of those with a short stature increased with a later birth order, significant differences in ORs were not observed between first-borns with and without siblings. In contrast, even after the adjustment of all covariates, second-borns and third-borns or more had 1.36- and 1.50-fold higher ORs of short stature at 4 years than first-borns with no siblings. In addition, we observed that sequentially adding covariates into Models 1 to 4 resulted in marginally increased ORs (Table 3).

### 3.4. Stratified Analyses by SGA, AGA, and LGA

Stratified analyses by birth size, such as SGA, AGA, and LGA, showed an independent association between birth order and short stature at 4 years of age. The analysis that included only the children born AGA did not materially change the overall results. The multiplicative interaction analysis confirmed that the associations between birth order and short stature at 4 years of age were not statistically affected by birth size (*p* for interaction >0.2) (Table 4). 

## 4. Discussion

This large-scale prospective birth cohort study in Japan found that the Z-score of birth length in later-borns was greater than that in first-borns, but later-borns were shorter than first-borns at 10 months, 1.5 years, 3 years, and 4 years. At the ages of 10 months, 1.5 years, 3 years, and 4 years, birth order was negatively associated with the Z-score of length/height. Second-borns and third-borns or more had significantly higher ORs of short stature at 4 years than first-borns with no siblings, while no differences were observed between the presence or absence of siblings for first-borns. The relationship was independent of birth weight, such as SGA, AGA, and LGA. To the best of our knowledge, this is the first study to show a relationship between birth order and the Z-score of length/height, including short stature, at 4 years of age.

The implications of the present findings are threefold. First, although the Z-score of birth length in later-borns was greater than that in first-borns, the Z-score of length in first-borns overtook that in later-borns by 10 months of age, and this relationship was maintained until 4 years of age. This finding is partly consistent with that in previous studies in Brazil [21] and the United Kingdom [22]. Wells at al. indicated that the Z-score of length in first-borns overtook that in later-borns at 6 months of age among 453 Brazilian children [21]. Ong et al. reported that infants of primiparous pregnancies were thin and short at birth, showed dramatic catch-up growth, and became taller than infants of non-primiparous pregnancies from 12 month of age among 1335 British children [22]. Our study adds evidence that, in Japan, the time point at which first-borns overtake later-borns is 10 months of age. However, the findings on differences in birth length between first- and later-borns are inconsistent among previous studies [21,22,24,40]. Our study indicates a significantly greater birth length in later-borns, similar to that reported by Ong et al. [21,22]. Further large-scale prospective studies in other regions are required to confirm this phenomenon. 

Second, the adjusted mean Z-score of length/height decreased gradually with a later birth order from 10 months to 4 years of age (*p* for linear trend <0.05). The degree of differences was narrow until 1.5 years of age, but it slightly widened at 3 and 4 years. These findings are consistent with those of previous reports. A Swedish large-scale cross-sectional study reported that second-, third-, and fourth-borns were 0.4, 0.7, and 0.8 cm significantly shorter than first-borns in adulthood [19]. A large-scale Dutch study also indicated that height in adulthood decreased by 0.1 to 0.4 cm for every one birth order later [18]. In a comparison of the degree of differences in the mean Z-score of height between first-borns with no siblings and second- and third-borns or more at 3 and 4 years of age, our study found that the differences ranged from 0.1 to 0.15 SD, which corresponds to approximately 0.4–0.6 cm. We considered that the degree of difference in height is also consistent with that in previous reports. This may suggest that the length/height differences that occur from infancy to early childhood are maintained unchanged into adulthood. A similar Brazilian study involving an analysis of variance from infancy to 4 years of age indicated that the length/height difference between 143 first-borns and 310 later-borns at 4 years of age was 0.4 SD [21], which is greater than that in our study. Although we could not determine the cause of this difference in SD, we believe that our study is larger and more accurate. In Table 2, the mean Z-scores of length/height are all negative from 10 months to 4 years of age in all groups. This is because the present study refers to data for Japanese children in 2000, which is consistent with the trend of a significant and gradual decline in the height of Japanese children over the past few decades [41].

Third, our study found that, even after the adjustment of all covariates and in the analysis limited to AGA, second-borns and third-borns or more had higher ORs of short stature at 4 years than first-borns with no siblings. This is a novel finding of the present study. Since the sequential adjustment of covariates in the models altered the values of ORs, the effects of potential confounding factors, such as maternal age at delivery, birth weight, and parents’ heights, make them strong confounders in the relationship between birth order and postnatal growth. However, the results of our study, which strictly controlled for potential confounders, suggest that a later birth order must be considered one of the factors predisposing to short stature. In Japan, growth hormone therapy is used for short stature in SGA children with insufficient catch-up growth after 3 years of age. In this study, the adjusted OR for short stature at 4 years of age was significantly higher in later-borns than in first-borns, even in the analysis limited to the SGA group. This result suggests that a more careful observation may be advisable in later-borns to avoid missing appropriate growth hormone therapeutic intervention in SGA children.

Pregnancies induce epigenetic changes in the mother, placenta, and later-borns, which affect the child’s metabolic regulation and appetite centers. In addition, deliveries affect pelvic morphology. It is assumed that these combined factors resulted in the current findings. The pelvic height and placenta are known to be larger in postpartum women than in primiparous women [42,43]. These may make later-borns less restricted in their growth in utero due to fewer structural restrictions and a more adequate supply of oxygen and nutrients [44]. Additionally, it has been reported that pregnancy affects DNA methylation in maternal blood and that the DNA methylation status of the placenta varies with the number of births [45,46], and these differences could also affect fetal growth. 

It is reasonable to assume that, in first-borns, who are more prone to fetal growth restriction than later-borns, epigenetic changes tend to be programmed to enhance hypothalamic appetite centers and suppress metabolic functions to ensure proper survival. For example, SGA-born infants are particularly strong cases of fetal growth restriction, and they are well known to experience greater adiposity and increased insulin resistance associated with weight catch-up in early childhood [47]. It is speculated that later-borns are less likely to experience these postnatal growth-promoting changes because they are less growth-restricted during fetal life than first-borns. Furthermore, in a prior experiment in sheep, mRNA expression in the liver at 30 days postpartum was lower for growth hormone receptors and IGF-1 receptors in infants from multiparous mothers than in those from nulliparous mothers, although IGF-1 was similar [48]. This may suggest that later-borns are less sensitive to growth hormones and IGF-1 than first-borns.

Several previous studies interpreted that the inverse association between birth order and postnatal growth caused by unfairly dividing family resources in later-borns compared with first-borns leads to infancy nutritional disadvantage [12,16,24]. However, in the present study, no significant length/height differences were demonstrated according to the presence or absence of siblings in first-borns. Although the results may be different in regions with more severe nutritional environments, we hypothesize that, at least in Japan, the presence of younger siblings during infancy and early childhood is not a negative factor in the nutritional environment after birth. 

A report in Nigeria found an OR of 7.331 (*p* < 0.001) for low birth weight when the inter-pregnancy interval was less than 18 months compared with when it was not [49]. Moreover, an Indian report has shown that the first-born height advantage disappears when later-born children are born at least 3 years after their elder siblings [16]. These could be considered indications that time is needed for maternal health to improve following childbirth. However, it may be noted that these results may differ from those in Japan, as the economic and nutritional conditions of the area are expected to be related to growth outcomes.

The strength of the present study is that it was a large-scale, nationwide prospective birth cohort research, which included important covariates, such as parents’ heights and birth size. Using this expansive dataset, we were able to report that second- and third-borns or more had higher ORs of short stature at 4 years than first-borns with no siblings. However, this study has some limitations. First, the participants had a significantly older maternal age at delivery and a higher proportion of first-borns than the original cohort. This may reflect the characteristics of the mothers or caregivers who reported the value of children’s length/height at 10 months, 1.5 years, 3 years, and 4 years of age. Second, the heights of the children and parents were self-reported; this may have created a response bias. Finally, birth spacing and underlying maternal diseases, such as gestational diabetes and pre-eclampsia, could not be adjusted for as potential confounders. Further large-scale studies are warranted to confirm our novel findings.

## 5. Conclusions

The Z-score of birth length in later-borns was greater than that in first-borns. However, later-borns were shorter than first-borns at 10 months, 1.5 years, 3 years, and 4 years; furthermore, birth order was negatively associated with length/height. This is the first study to report that second-borns and third-borns or more had higher ORs of short stature at 4 years than first-borns with no siblings, regardless of birth size. Birth order may be considered one of the factors that predispose to short stature in early childhood.

## Figures and Tables

**Figure 1 children-10-00557-f001:**
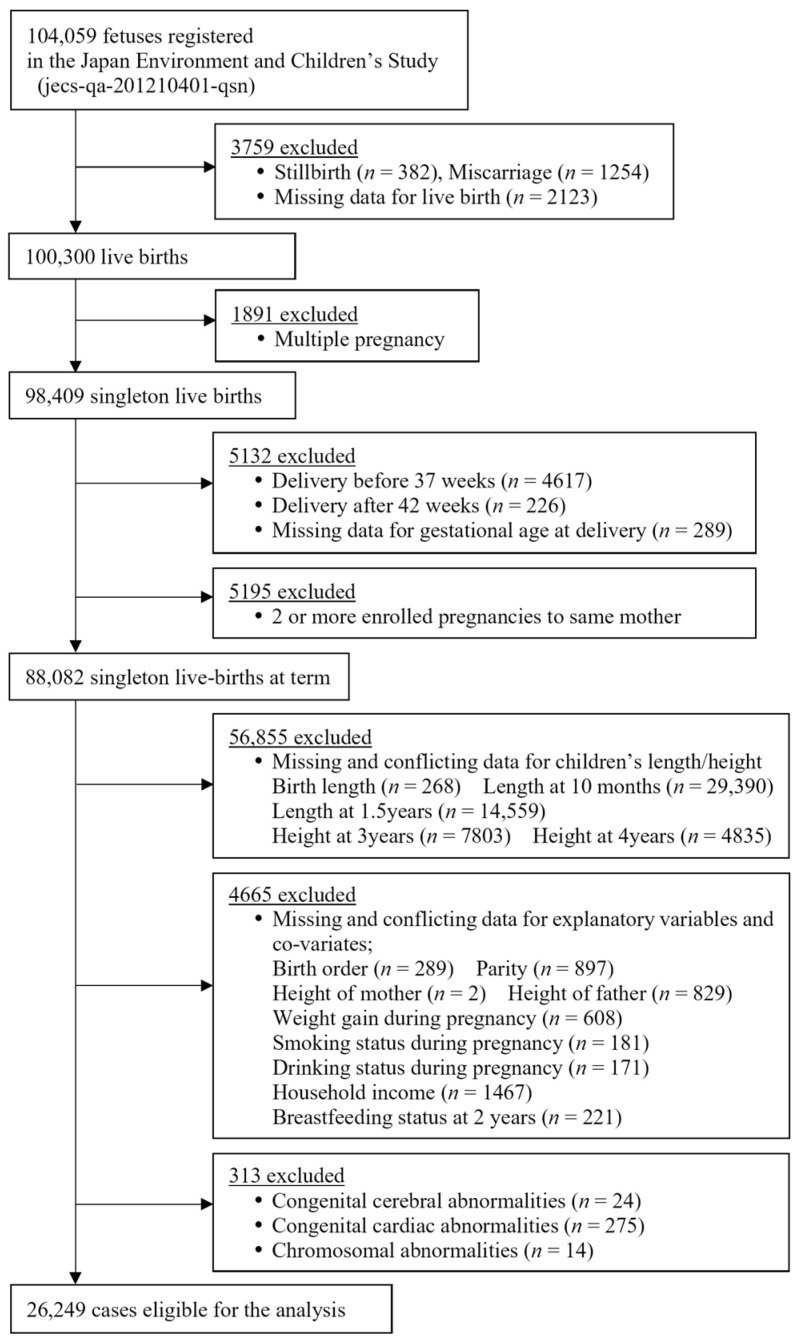
Study enrollment flowchart.

**Table 1 children-10-00557-t001:** Participant characteristics by birth order (*n* = 26,249).

	First-Borns, Sibling (−)	First-Borns, Sibling (+)	Second-Borns	Third-Borns or More
	(*n* = 9657)	(*n* = 3463)	(*n* = 9240)	(*n* = 3889)
Sex of child, *n* (%)				
Male	4804 (49.7)	1800 (52.0)	4757 (51.5)	1959 (50.4)
Female	4853 (50.3)	1663 (48.0)	4483 (48.5)	1930 (49.6)
Z-score of birth weight	0.08 ± 0.98	0.09 ± 0.99	0.03 ± 0.91	0.10 ± 0.96
Age of mother at delivery (years)	31.68 ± 4.88	29.43 ± 4.21	32.66 ± 4.30	33.78 ± 3.94
Maternal weight gain during pregnancy (kg)	10.39 ± 3.78	10.69 ± 3.69	9.86 ± 3.58	9.92 ± 3.55
Smoking status during pregnancy, *n* (%)				
Never smoked	6347 (65.7)	2311 (66.7)	5853 (63.3)	2364 (60.8)
Quit smoking before recognizing pregnancy	1904 (19.7)	624 (18.0)	2490 (26.9)	1078 (27.7)
Quit smoking after finding out pregnancy	1231 (12.7)	452 (13.1)	714 (7.7)	313 (8.0)
Currently smoking	175 (1.8)	76 (2.2)	183 (2.0)	134 (3.4)
Drinking status during pregnancy, *n* (%)				
Never drinker	3042 (31.5)	1063 (30.7)	3355 (36.3)	1409 (36.2)
Former drinker	6485 (67.2)	2354 (68.0)	5624 (60.9)	2276 (58.5)
Current drinker	130 (1.3)	46 (1.3)	261 (2.8)	204 (5.2)
Household income, *n* (%)				
<2 million JPY	285 (3.0)	121 (3.5)	255 (2.8)	147 (3.8)
2–5 million JPY	6187 (64.1)	2241 (64.7)	6159 (66.7)	2603 (66.9)
6–10 million JPY	2725 (28.2)	928 (26.8)	2433 (26.3)	961 (24.7)
≥10	460 (4.8)	173 (5.0)	393 (4.3)	178 (4.6)
Breastfeeding status at 2 yr, *n* (%)				
Currently breastfeeding	1797 (18.6)	88 (2.5)	1842 (19.9)	947 (24.4)
Stopped breastfeeding before	6934 (71.8)	3086 (89.1)	6705 (72.6)	2595 (66.7)
Never breastfed	926 (9.6)	289 (8.3)	693 (7.5)	347 (8.9)
Height of mother (cm)	158.33 ± 5.32	158.41 ± 5.24	158.35 ± 5.27	158.16 ± 5.22
Height of father (cm)	171.90 ± 5.77	171.97 ± 5.81	172.10 ± 5.77	171.74 ± 5.67

**Table 2 children-10-00557-t002:** Adjusted means and 95% confidence intervals of Z-score of height from birth to 4 years of age according to birth order (*n* = 26,249).

Age	Group IA: First-Borns, Sibling (−)	Group IB: First-Borns, Sibling (+)	Group II: Second-Borns	Group III: Third-Borns or More	*p* for Linear Trend †	*p* for Post Hoc Test Using Bonferroni					
						IA vs. IB	IA vs. II	IA vs. III	IB vs. II	IB vs. III	II vs. III
Birth	−0.026 (−0.041; −0.011)	−0.013 (−0.039; 0.013)	0.136 (0.120; 0.151)	0.120 (0.097; 0.144)	<0.01	0.99	<0.01	<0.01	<0.01	<0.01	0.99
10 months	−0.324 (−0.343; −0.304)	−0.330 (−0.364; −0.297)	−0.466 (−0.486; −0.446)	−0.569 (−0.600; −0.538)	<0.01	0.99	<0.01	<0.01	<0.01	<0.01	<0.01
1.5 years	−0.558 (−0.580; −0.537)	−0.531 (−0.568; −0.494)	−0.642 (−0.664; −0.620)	−0.658 (−0.692; −0.623)	<0.01	0.99	<0.01	<0.01	<0.01	<0.01	0.99
3 years	−0.171 (−0.188; −0.154)	−0.166 (−0.195; −0.137)	−0.295 (−0.312; −0.277)	−0.340 (−0.367; −0.312)	<0.01	0.99	<0.01	<0.01	<0.01	<0.01	<0.05
4 years	−0.156 (−0.173; −0.140)	−0.182 (−0.210; −0.153)	−0.267 (−0.284; −0.250)	−0.323 (−0.350; −0.296)	<0.01	0.82	<0.01	<0.01	<0.01	<0.01	<0.01

Adjusted for age of mother at delivery (years), Z-score of birth weight, parents’ heights (cm), maternal weight gain during pregnancy (kg), smoking status during pregnancy (Never; Previously did, but quit before recognizing current pregnancy; Previously did, but quit after finding out current pregnancy; Yes, I still smoke), alcohol drinking status during pregnancy (non-drinkers, former drinkers, current drinkers), household income (<2, 2–5, 6–9, ≥10 million JPY), breastfeeding status at 2 years (currently breastfeeding, stopped breastfeeding before, never breastfed), and Study Areas (Hokkaido, Miyagi, Fukushima, Chiba, Kanagawa, Koshin, Toyama, Aichi, Kyoto, Osaka, Hyogo, Tottori, Kochi, Fukuoka, and South Kyushu/Okinawa). † Polynomial method in the general linear model.

**Table 3 children-10-00557-t003:** Odds ratios and 95% confidence intervals for short stature at 4 years of age according to birth order (*n* = 26,249).

	First-Borns, Sibling (−)	First-Borns, Sibling (+)	Second-Borns	Third-Borns or More	*p* for Linear Trend
	(*n* = 9657)	(*n* = 3463)	(*n* = 9240)	(*n* = 3889)	
Short stature at 4 yr, *n* (%)	255 (2.6)	96 (2.8)	301 (3.3)	140 (3.6)	
Crude odds ratio	Reference	1.05 (0.83; 1.33)	1.24 (1.05; 1.47)	1.38 (1.12; 1.70)	<0.01
Model 1		0.99 (0.78; 1.26)	1.28 (1.08; 1.52)	1.46 (1.18; 1.81)	<0.01
Model 2		0.99 (0.78; 1.26)	1.27 (1.07; 1.51)	1.52 (1.23; 1.89)	<0.01
Model 3		1.03 (0.80; 1.33)	1.34 (1.12; 1.60)	1.49 (1.20; 1.86)	<0.01
Model 4		1.08 (0.84; 1.39)	1.36 (1.13; 1.62)	1.50 (1.20; 1.88)	<0.01

Model 1: adjusted for age of mother at delivery (years). Model 2: adjusted for age of mother at delivery (years) and Z-score of weight at birth. Model 3: adjusted for variables in Model 2 + parent’s heights. Model 4: adjusted for variables in Model 3 + maternal weight gain during pregnancy (kg), smoking status during pregnancy (Never; Previously did, but quit before recognizing current pregnancy; Previously did, but quit after finding out current pregnancy; Yes, I still smoke), alcohol drinking status during pregnancy (non-drinkers, former drinkers, current drinkers), household income (<2, 2–5, 6–9, ≥10 million JPY), breastfeeding status at 2 years (currently breastfeeding, stopped breastfeeding before, never breastfed), and Study Areas (Hokkaido, Miyagi, Fukushima, Chiba, Kanagawa, Koshin, Toyama, Aichi, Kyoto, Osaka, Hyogo, Tottori, Kochi, Fukuoka, and South Kyushu/Okinawa).

**Table 4 children-10-00557-t004:** Odds ratios and 95% confidence intervals for short stature at 4 years of age according to birth order stratified by birth size for gestational age.

	First-Borns, Sibling (−)	First-Borns, Sibling (+)	Second-Borns	Third-Borns or More	*p* for Linear Trend	*p* for Interaction
Small for gestational age (*n* = 1915)						0.62
	(*n* = 732)	(*n* = 241)	(*n* = 659)	(*n* = 283)		
Short stature at 4 yr, *n* (%)	63 (8.6)	24 (10.0)	67 (10.2)	27 (9.5)		
Crude odds ratio	Reference	1.17 (0.72; 1.93)	1.20 (0.84; 1.73)	1.12 (0.70; 1.80)	0.43	
Adjusted odds ratio		1.17 (0.68; 2.03)	1.63 (1.09; 2.43)	1.57 (0.93; 2.66)	<0.05	
Appropriate for gestational age (*n* = 21,717)						
	(*n* = 7905)	(*n* = 2840)	(*n* = 7789)	(*n* = 3183)		
Short stature at 4-yr, *n* (%)	186 (2.4)	69 (2.4)	225 (2.9)	111 (3.5)		
Crude odds ratio	Reference	1.03 (0.78; 1.37)	1.23 (1.01;1.50)	1.50 (1.18; 1.90)	<0.01	
Adjusted odds ratio		1.02 (0.76; 1.37)	1.30 (1.05; 1.60)	1.54 (1.20; 1.99)	<0.01	
Large for gestational age (*n* = 2917)						
	(*n* = 1020)	(*n* = 382)	(*n* = 792)	(*n* = 423)		
Short stature at 4 yr, *n* (%)	6 (0.6)	3 (0.8)	9 (1.1)	2 (0.5)		
Crude odds ratio	Reference	1.34 (0.33; 5.38)	1.94 (0.69; 5.48)	0.80 (0.16; 3.99)	0.65	
Adjusted odds ratio		1.99 (0.45; 8.95)	2.01 (0.68; 6.00)	1.09 (0.21; 5.77)	0.46	

Adjusted for age of mother at delivery (years), Z-score of weight at birth, maternal weight gain during pregnancy (kg), smoking status during pregnancy (Never; Previously did, but quit before recognizing current pregnancy; Previously did, but quit after finding out current pregnancy; Yes, I still smoke), alcohol drinking status during pregnancy (non-drinkers, former drinkers, current drinkers), household income (<2, 2–5, 6–9, ≥10 million JPY), breastfeeding status at 2 years (currently breastfeeding, stopped breastfeeding before, never breastfed), parent’s heights, and Study Areas (Hokkaido, Miyagi, Fukushima, Chiba, Kanagawa, Koshin, Toyama, Aichi, Kyoto, Osaka, Hyogo, Tottori, Kochi, Fukuoka, and South Kyushu/Okinawa).

## Data Availability

The data are unsuitable for public deposition due to ethical restrictions and the legal framework in Japan. It is prohibited by the Act on the Protection of Personal Information (Act No. 57 of 30 May 2003, amendment on 9 September 2015) to publicly deposit data containing personal information. The Ethical Guidelines for Medical and Health Research Involving Human Subjects enforced by the Japan Ministry of Education, Culture, Sports, Science and Technology and the Ministry of Health, Labour and Welfare also restrict the open sharing of epidemiologic data. All inquiries about access to data should be sent to jecs-en@nies.go.jp. The person responsible for handling inquiries sent to this e-mail address is Dr. Shoji F. Nakayama, JECS Programme Office, National Institute for Environmental Studies.

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
