# Peer review of "Relationship between Birth Order and Postnatal Growth until 4 Years of Age: The Japan Environment and Children’s Study"

_children, 2023, doi:10.3390/children10030557_

Round 1

Reviewer 1 Report

Interesting paper based on a large study group

However, on the basis of the obtained results, it is difficult to conclude that birth order can lead to short stature, maybe predispose would be a more appropriate term.

In line 388 and 426 I would suggest to use the word "predispose" instead of "lead"

Misspellings:

line 170 - lambda

line 252 - maternal

Author Response

Interesting paper based on a large study group However, on the basis of the obtained results, it is difficult to conclude that birth order can lead to short stature, maybe predispose would be a more appropriate term. In line 388 and 426 I would suggest to use the word "predispose" instead of "lead".

[Authors’ response]

According to the suggestion, we have used the word "predispose" instead of "lead".

[Changes in the revised manuscript]

(Page 1, Line 31-32)

Birth order was significantly associated with postnatal growth and may be a factor predisposing to short stature in early childhood.

(Page 10-11, Line 400-402)

However, the results of our study, which strictly controlled potential confounders, suggest that later birth order must be considered one of the factors predisposing to short stature.

(Page 12, Line 464-465)

Birth order may be considered one of the factors that predispose to short stature in early childhood.

[Reviewer 1’s comment #2]

Misspellings: line 170 - lambda, line 252 - maternal

[Authors’ response]

We apologize for the errors and have corrected them.

[Changes in the revised manuscript]

(Page 5, Line 179-182)

The Z-scores of length/height were calculated according to the growth standard charts for Japanese children using the LMS (lambda-mu-sigma) method, with the closest LMS values for each measurement age used [29,30]. Short stature at 4 years of age was defined as less than -2 SD.

(Page 6, Line 262-263)

Maternal weight gain during pregnancy was higher in the group of first-borns compared with others.

Reviewer 2 Report

This study investigated the relationship between birth order and birth length and child height, up to 4 years of age.  The experimental design included a comparison between first-borns with and without siblings (although the number of siblings was not included).  The findings have confirmed those observed previously in other cohorts globally.

1. Abstract - include comments on the potential confounding factors that were considered in the analysis, such as sex of the child.

2. Line 65 - add the study hypothesis.

3. Line 107-109 - were the data significantly different?  If so, include the p-values. 

4. Did the study consider the effect of birth spacing on the data?  Also, were all the individuals from separate families?

5. Line 252 - correct the spelling of 'maternal'.

6. Indicate the statistical effects in Table 1 and avoid the use of 'slightly higher/lower' in the text that relates to this table.

7. In Table 2, the z-scores all appear lower than the standards - why is this?

8. Line 358 - it is not possible to state that the change occurred at 10 months specifically, only 'by 10 months'.  Earlier time points were not investigated.

9. The Discussion could include more about the possible mechanisms for the findings.  Expand on the evidence for differences in placental structure and function, epigenetics and fetal programming of appetite.

10. Do the authors have any information about the relationship between birth order and body weight or BMI in this cohort?  This would be an important addition to the study with relevance for the risk of obesity.

Author Response

This study investigated the relationship between birth order and birth length and child height, up to 4 years of age.  The experimental design included a comparison between first-borns with and without siblings (although the number of siblings was not included).  The findings have confirmed those observed previously in other cohorts globally.

Abstract - include comments on the potential confounding factors that were considered in the analysis, such as sex of the child.

[Authors’ response]

We have enumerated the potential confounding factors in the Abstract and reduced several repeated words to adjust the number of words.

[Changes in the revised manuscript]

(Page 1, Line 22-25)

General linear and multivariable logistic regression models were performed and adjusted for birth weight, parents’ heights, maternal age at delivery, gestational weight gain, maternal smoking and alcohol drinking status during pregnancy, household income, breastfeeding status, and Study Areas.

[Reviewer 2’s comment #2]

Line 65 - add the study hypothesis.

[Authors’ response]

We have added the study hypothesis in the Introduction as per your suggestion.

[Changes in the revised manuscript]

(Page 2, Line 63-69)

Therefore, this study aimed to investigate the hypothesis that, even in Japan, later-borns are born larger than first-borns, but growth reverses during early childhood, which may be a risk factor for short stature. To this end, we examined the relationship between birth order (considering existence of a sibling for the first-borns) and postnatal growth using Z-score of height at birth, 10 months, 1.5 years, 3 years, and 4 years of age and the association between birth order and short stature among 4-year-old children, using large data sets from ongoing multicentre prospective birth cohort studies across Japan.

[Reviewer 2’s comment #3]

Line 107-109 - were the data significantly different?  If so, include the p-values.

[Authors’ response]

According to your suggestion, we have added the p-value in the manuscript while avoiding use of the word “slightly”. Furthermore, related text was also revised such as the Limitations in the Discussion.

[Changes in the revised manuscript]

(Page 3, Line 118-120)

However, the maternal age at delivery and the proportion of first-borns in the current analysis were significantly higher than among the overall JECS participants with singleton live births (p<0.05, Results not shown).

(Page 11, Line 450-452)

First, the participants had significantly older maternal age at delivery and higher proportion of first-borns compared with the original cohort.

[Reviewer 2’s comment #4]

Did the study consider the effect of birth spacing on the data? Also, were all the individuals from separate families?

[Authors’ response]

We do consider “birth spacing” an important confounder; however, the JECS dataset has no clear information to identify birth spacing. We have added this point in the Discussion and presented it as a limitation. In addition, all study participants were individuals from separate families. Although this point was mentioned in Figure 1, we have also added it in the text for better clarity.

[Changes in the revised manuscript]

(Page 3, Line 101-102)

two or more enrolled pregnancies to the same mother (n=5,195) were excluded

(Page 11, Line 438-445)

A report in Nigeria found an OR of 7.331 (P < 0.001) for low birth weight when the inter-pregnancy interval was less than 18 months compared with when it was not [49]. Moreover, an Indian report has shown that the firstborn height advantage disappears when later-born children are born at least 3 years after their elder siblings [16]. These could be considered indications that time is needed for maternal health to improve following childbirth. However, it may be noted that in actual, these results may differ from those in Japan, as the economic and nutritional conditions of the area are expected to be related to growth outcomes.

(Page 11, Line 455-456)

Finally, birth spacing and underlying maternal diseases such as gestational diabetes and pre-eclampsia could not be adjusted for as potential confounders.

[Reviewer 2’s comment #5]

Line 252 - correct the spelling of 'maternal'.

[Authors’ response]

We have corrected it.

[Changes in the revised manuscript]

(Page 6, Line 262-263)

Maternal weight gain during pregnancy was higher in the group of first-borns compared with others.

[Reviewer 2’s comment #6]

Indicate the statistical effects in Table 1 and avoid the use of 'slightly higher/lower' in the text that relates to this table.

[Authors’ response]

Since, JECS rules recommend the presentation of p-values in descriptive tables be avoided according to the Strengthening the Reporting of Observational Studies in Epidemiology (STROBE) and the ASA Statement††, we were not able to show the statistical effects in Table 1. However, to avoid the expression which you have pointed out (‘slightly higher/lower’), we have revised the first paragraph in the Results. If the p-value in descriptive tables must be presented in order to be published in the journal, we will try to convince JECS statistical office.

† Vandenbroucke JP, von Elm E, Altman DG, Gøtzsche PC, Mulrow CD, Pocock SJ, Poole C, Schlesselman JJ, Egger M; STROBE Initiative. Strengthening the Reporting of Observational Studies in Epidemiology (STROBE): explanation and elaboration. PLoS Med. 2007 Oct 16;4(10):e297. doi: 10.1371/journal.pmed.0040297.

†† Wasserstein RL, Lazar NA. The ASA’s statement on p-values: context, process, and purpose. Am Statis 2016.

[Changes in the revised manuscript]

(Page 6, Line 268-269)

Significantly higher paternal height was observed in the group of second borns (Table 1).

[Reviewer 2’s comment #7]

In Table 2, the z-scores all appear lower than the standards - why is this?

[Authors’ response]

We have added the reasons that our results showed lower z-scores of height than the standards in the Discussion.

[Changes in the revised manuscript]

(Page 10, Line 389-393)

In Table 2, the mean Z-scores for length/height are all negative from 10 months to 4 years of age in all groups. This is because the present study refers to data for Japanese children in 2000, which is consistent with the trend of a significant and gradual decline in the height of Japanese children over the past few decades [41].

[Reviewer 2’s comment #8]

Line 358 - it is not possible to state that the change occurred at 10 months specifically, only 'by 10 months'.  Earlier time points were not investigated.

[Authors’ response]

According to your suggestion, we have changed the expression.

[Changes in the revised manuscript]

(Page 10, Line 367-368)

Our study added evidence that, in Japan, the time point at which first-borns overtake later-borns is by 10 months of age.

[Reviewer 2’s comment #9]

The Discussion could include more about the possible mechanisms for the findings.  Expand on the evidence for differences in placental structure and function, epigenetics and fetal programming of appetite.

[Authors’ response]

We have added the possible mechanisms for these findings.

[Changes in the revised manuscript]

(Page 11, Line 408-429)

Pregnancies induce epigenetic changes in the mother, placenta and later-borns, which affect the child's metabolic regulation and appetite centers. In addition, deliveries affect pelvic morphology. It is assumed that these combined factors have resulted in the current findings. The pelvic height and placenta are known to be larger in postpartum women compared with primiparous women [42,43]. These may make later-borns less restricted in their growth in utero due to fewer structural restrictions and more adequate supply of oxygen and nutrients [44]. Additionally, it has been reported that pregnancy affects DNA methylation in maternal blood and that the DNA methylation status of the placenta varies with the number of births [45,46], and these differences could also affect fetal growth.

It is reasonable to assume that in first-borns, who are more prone to fetal growth re-striction than later-borns, epigenetic changes tend to be programmed to enhance hypo-thalamic appetite centers and suppress metabolic functions to ensure proper survival. For example, SGA born infants are particularly strong cases of fetal growth restriction, and they are well known to experience greater adiposity and increased insulin resistance as-sociated with weight catch-up in early childhood [47]. It is speculated that later-borns are less likely to experience these postnatal growth-promoting changes because they are less growth-restricted during fetal life than first-borns. Furthermore, in a prior experiment in sheep, mRNA expression in the liver at 30 days postpartum was lower for growth hormone receptors and IGF-1 receptors in infants from multiparous mothers than in those from nulliparous mothers, although IGF-1 was similar [48]. This may suggest that later-borns are less sensitive to growth hormone and IGF-1 compared with first-borns.

[Reviewer 2’s comment #10]

Do the authors have any information about the relationship between birth order and body weight or BMI in this cohort?  This would be an important addition to the study with relevance for the risk of obesity.

[Authors’ response]

We also believe that it is very important to study the relationship between birth order and weight or BMI. However, the JECS data analysis did not allow us to detect a trend in these relationships. This is possibly because weight and BMI are more strongly influenced by factors of physical activity and diet than by height. After discussion between the co-authors, we have decided not to include information on weight and BMI in order to avoid complications with the study content and discussion, as the present study focused on height.

Reviewer 3 Report

Gestational diseases (gestational diabetes, pre-eclampsia for example) were not considered and can affect the gestational outcomes ans ghild growth.

Twin pregnancy and delivery time was declared in “multiple pregnancy in Figure 1” or in other sections but coud be included in description because is na importante and classical factor acting in child growth.

Model 4 of multivariable adjusted logistic regression included a lot os factors and variables that need to evaluated separately. For example, weight gain during pregnancy is not clear that afects birth weight in small or great to gestational age while maternal smoking and drinking during pergnancy thens affect decreased the birthweight but can lead to cath-up growth, between others.

Line 252: typing error (“Maternaa”)

Author Response

Gestational diseases (gestational diabetes, pre-eclampsia for example) were not considered and can affect the gestational outcomes ans ghild growth.

[Authors’ response]

Although questions were asked about gestational diseases, unfortunately, this study was not able to investigate the association you have mentioned. Hence, the following line has been added to the Limitations.

[Changes in the revised manuscript]

(Page 12, Line 455-456)

Finally, birth spacing and underlying maternal diseases such as gestational diabetes and pre-eclampsia could not be adjusted for as potential confounders.

[Reviewer 3’s comment #2]

Twin pregnancy and delivery time was declared in “multiple pregnancy in Figure 1” or in other sections but coud be included in description because is na importante and classical factor acting in child growth.

[Authors’ response]

Multiple births are excluded in this study because they are known to be growth suppressed compared with singleton births. We have added the following sentences in the Abstract and Methods for better clarity.

[Changes in the revised manuscript]

(Page 1, Line 21)

A total of 26,249 full-term singleton births were targeted.

(Page 3, Line 98-99)

Multiple births, preterm, and post-term infants were excluded because their growth patterns differ from those of term singletons [27,28].

[Reviewer 3’s comment #3]

Model 4 of multivariable adjusted logistic regression included a lot os factors and variables that need to evaluated separately. For example, weight gain during pregnancy is not clear that pregnan birth weight in small or great to gestational age while maternal smoking and drinking during pregnancy thens affect decreased the birthweight but can lead to cath-up growth, between others.

[Authors’ response]

We investigated the association between each confounder and short stature at 4 years of age to determine the final model for regression. For example, a 1 kg increase in weight during pregnancy showed a significantly lower odds ratio 0.95 (95% confidence intervals: 0.94-0.97) for short stature in 4 year-old children. Hence, we determined the covariates for Model 4 and made sure that the fully adjusted model reasonably well discriminated short stature at 4 years of age, with a rate of correct classification of 90.2% and a p-value of 0.55 for goodness-of-fit by the Hosmer-Lemeshow test.

[Reviewer 3’s comment #4]

Line 252: typing error (“Maternaa”)

[Authors’ response]

We apologize for the error and have corrected it.

[Changes in the revised manuscript]

(Page 6, Line 262-263)

Maternal weight gain during pregnancy was higher in the group of first-borns compared with others.

Thank you for your consideration. I look forward to hearing from you.

Round 2

Reviewer 2 Report

The authors have addressed my comments satisfactorily.  It would be useful to identify statistical differences in Table 1, but I am happy to follow the journal format.